

# Identification of genes involved in the evolution of human intelligence through combination of inter-species and intra-species genetic variations

Mengjie Li, Wenting Zhang and Xiaoyi Zhou

College of Life Sciences, Shanghai Normal University, Shanghai, China

## ABSTRACT

Understanding the evolution of human intelligence is an important undertaking in the science of human genetics. A great deal of biological research has been conducted to search for genes which are related to the significant increase in human brain volume and cerebral cortex complexity during hominid evolution. However, genetic changes affecting intelligence in hominid evolution have remained elusive. We supposed that a subset of intelligence-related genes, which harbored intra-species variations in human populations, may also be evolution-related genes which harbored inter-species variations between humans (*Homo sapiens*) and great apes (including *Pan troglodytes* and *Pongo abelii*). Here we combined inter-species and intra-species genetic variations to discover genes involved in the evolution of human intelligence. Information was collected from published GWAS works on intelligence and a total of 549 genes located within the intelligence-associated loci were identified. The intelligence-related genes containing human-specific variations were detected based on the latest high-quality genome assemblies of three human's closest species. Finally, we identified 40 strong candidates involved in human intelligence evolution. Expression analysis using RNA-Seq data revealed that most of the genes displayed a relatively high expression in the cerebral cortex. For these genes, there is a distinct expression pattern between humans and other species, especially in neocortex tissues. Our work provided a list of strong candidates for the evolution of human intelligence, and also implied that some intelligence-related genes may undergo inter-species evolution and contain intra-species variation.

## INTRODUCTION

Humans have brains with significantly increased size and complexity compared to their ape counterparts (*Rakic, 2009*; *Chenn & Walsh, 2002*; *Lui, Hansen & Kriegstein, 2011*). Corresponding alterations in intelligence have helped humans survive and create tools (*Deary, 2012*). Inspection of human genomic differences from our closest evolutionary relatives could help us to understand the intelligence-related genetic events during hominid evolution. The intelligence difference is thought to be derived from changes in genetics,

Corresponding author
Xiaoyi Zhou, zhouxy@shnu.edu.cn

owing to a small fraction of the 1% of sequence differences between the human genome and the chimpanzee genome, in which the human-specific gene insertions, deletions, and duplications played a critical role (*Cheng et al., 2005*).

Various approaches in molecular biology have been used to search for the human-specific genes and mutations therein that were related to the human intelligence. Several candidate genes involved in human intelligence evolution were identified based on human-ape comparative genomic analysis, gene expression profiling and other functional evidences. For example, in a very recent study, the information from gene expression profiling was integrated with the information from gene duplications in the ape and human lineages, which was then used to search for the human-specific genes that were highly expressed during human corticogenesis. In > 35 candidates obtained through bioinformatics analysis, *NOTCH2NL* was functionally investigated—it was found that the gene was able to promote the expansion of cortical progenitors, serving as an important gene contributing to the evolution of the human brain (*Fiddes et al., 2018*; *Suzuki et al., 2018*). More recently, several human-specific genes have been identified and the critical genetic changes often occurred in gene regulation regions or resulted from the human-specific gene duplications; these include the *NOTCH2NL* gene, as well as *FZD8*, *SRGAP2*, *ARHGAP11B*, and *TBC1D3*, (*Boyd et al., 2015*; *Dennis et al., 2012*; *Charrier et al., 2012*; *Florio et al., 2015*; *Ju et al., 2016*).

As a highly heritable trait, intelligence has been intensively investigated using forward genetic approaches (*Davies et al., 2015*; *Davies et al., 2016*; *Sniekers et al., 2017*; *Trampush et al., 2017*; *Zabaneh et al., 2018*; *Savage et al., 2018*; *Davies et al., 2018*; *Hill et al., 2016*; *Hill et al., 2019*). Several genome-wide association studies (GWAS) and meta-analyses using very large human populations have been performed to identify causative genomic loci and genes underlying intelligence. Despite a significant enrichment in the nervous system, the functional links of the identified genes are diverse in which a wide variety of genes are involved (*Davies et al., 2015*; *Davies et al., 2016*; *Sniekers et al., 2017*; *Trampush et al., 2017*; *Zabaneh et al., 2018*; *Savage et al., 2018*). This suggests that the evolution of human intelligence is a complicated process. While most causative genes may directly affect the central nervous system, genes from many related biological processes may be involved in the intelligence evolution as well.

Investigating the human-specific variations (genetic differences between humans and great apes) could provide key clues for understanding the process of the evolution of human intelligence. As the assembling gaps and errors in the previous reference genomes of the great apes (e.g., the human sequence guided assembling from short reads), the previous genomes were not qualified enough for the detection of complex structural variations (*Prufer et al., 2012*; *Scally et al., 2012*; *Prado-Martinez et al., 2013*) such as tandem repeats, large-scale inversions, and duplications. These structural variations usually play important roles in human evolution (*McLean et al., 2011*). Hence, comparative genomic analysis from the complete genome sequences of both humans and great apes is needed to comprehensively identify the genetic variation. Recently, the high-quality genome sequences of three of human's closest relatives, chimpanzee, orangutan, and gorilla, were generated from long-read sequencing (PacBio technology) and de novo assembling (*Kronenberg et al.,*

*2018*; *Gordon et al., 2016*). The chromosome-level contiguous genome assemblies facilitate a deeper understanding of the genomic differences between these species.

Among a number of genomic differences between humans and apes, it is very important but technically difficult to know which variants are specific to intelligence. In order to further search out the candidate genes related to the evolution of human intelligence, we suppose that some intelligence genes may have both inter-species (between humans and the great apes) and intra-species (within humans) variations. For intra-species variations, we collected genomic loci identified by several sets of GWAS on human intelligence, and genes in these loci (termed as intelligence-associated-genes) could be regarded to contain intra-species intelligence differences. For inter-species variation, genomic differences between humans and the great apes, revealed by recent high-quality sequencing (*Kronenberg et al., 2018*; *Gordon et al., 2016*), were used and filtered. Hence, intelligence evolution was integrated by the overlap of intelligence-associated genes and human-specific variations. We found that many of the intelligence-associated genes, including tens of strong candidate genes related to the evolution of human intelligence, contained human-specific structural variations. Coupled with the expression profiling of the genes, this genome-wide analysis provided a useful resource for the evolutionary genetic studies on intelligence.

## MATERIALS & METHODS

### Preparation of bait genes

To exploit the genes related to human intelligence, six major works were collected from GWAS or GWAS meta-analyses on human intelligence with a large sample size from the last five years (various intelligence-related phenotypes including general cognitive, reaction time and verbal-numerical reasoning). A total of 271 loci associated with intelligence in the human genome (Table S1) were identified in these six studies.

The six GWAS studies included: (i) meta-analyses of general cognitive function ($n = 53949$, *Davies et al., 2015*); (ii) GWAS of cognitive function and educational attainment ($n = 112151$, *Davies et al., 2016*); (iii) meta-analyses of calculated Spearman's $g$ or a primary measure of fluid intelligence ($n = 78308$, *Sniekers et al., 2017*); (iv) meta-analysis and gene-based analysis of human cognition using 24 cohorts ($n = 35298$, *Trampush et al., 2017*); (v) GWAS using human populations with extremely high intelligence ($n = 1238$, *Zabaneh et al., 2018*); (vi) a recent meta-analysis of 14 independent epidemiological cohorts with intelligence assessed ($n = 269867$, *Savage et al., 2018*).

All the independent, significantly associated SNPs (IndSigSNPs) nearest genes (based on ANNOVAR annotations) were integrated with redundancies (the same gene identified in more than one study) removed. We took these 549 genes as "bait genes" (Table S2), which were candidates for human intelligence-related genes.

It should be noted that some GWAS works underlying human intelligence that have been published very recently may be not included in this study. This would not affect our analyses because this study aimed to provide some candidate genes for human intelligence evolution and could not identify all related genes at one time. In addition, because genotype imputation was not performed for the X chromosome in some cohorts (e.g., the UKB

cohort including 195,653 samples with the assessed phenotype verbal and mathematical reasoning), the potential genes related to intelligence in the X chromosome were not included in our "bait genes".

## Preparation of variation pond

The new genome assemblies of chimpanzees, gorillas and orangutans have become fully available recently through single-molecule, real-time (SMRT) long-read sequencing, which improved the resolution of large and complex genomic regions. We incorporated the human-specific structural variations (from intermediate size to large size) into a "variation pond", including exon gain/loss, short tandem repeats (STRs), insertion/deletion (indels) of more than 50 bp, and inversions. Moreover, considering the important role of the human-specific segmental duplications (HSDs, > 1 kb sequence with > 90% similarity, indicating the large recent duplication events, *Bailey et al., 2001*) in new gene function and human evolution (*Dennis & Eichler, 2016*), the HSDs identified recently from the comparative genomic information of both macaque and mouse (*Dennis et al., 2017*) were added into the "pond". The "baits" and "ponds" were then integrated to detect whether there were human-specific variations hit by "bait genes", thus generating the "prey genes" (Tables S3–S5), which refer to the intelligence-related genes with human-specific variations. The human-specific variation that was located within each of the 549 genes was left for further analysis using a window of 1 Kb (the locations of the variations and genes on the chromosomes were divided by 1000 for the computation efficiency).

The genome sequences of three great apes were downloaded from the NCBI database. The reference genome sequences from *Pan troglodytes* (chimpanzee), *Pongo abelii* (Orangutan) and Gorilla (Western Lowland Gorilla) were downloaded from: https://www.ncbi.nlm.nih.gov/assembly/GCF_002880755.1/, https://www.ncbi.nlm.nih.gov/assembly/GCF_002880775.1/ and https://www.ncbi.nlm.nih.gov/assembly/GCF_000151905.2/, respectively.

Localized alignments of the target gene sequences were performed to filter out the false positive human-specific variations from previous reports (*McLean et al., 2011*; *Kronenberg et al., 2018*; *Dennis et al., 2017*). The local sequences of the human genome were retrieved from chromosomes found in the GRCh38 version, and were then aligned with the great ape genomes using BLASTn (ncbi-blast-2.2.28+ version) with the parameters "-evalue 1E-50 -dust no". The human-specific variation that was undetectable with a local BLAST was then removed for subsequent analyses.

## Investigation of HIEGs

Most "prey" genes contained the human-specific variation in introns, far away from the exon-intron junction site. These variants are less likely to affect the gene functions, so only genes that contained variations in the coding regions were considered, named human intelligence evolution genes (HIEGs). These HIEGs included all the genes containing exon-gain/loss (2 genes), hCONDEL (28 genes), or HSD (1 gene), and genes with exon-located indels (8 genes) or STRs (4 genes). 40 non-redundant genes were finally identified, which were associated with human intelligence and which carried significant human-specific variations.

The gene transcript information was obtained from Ensembl Release 95 (http://asia.ensembl.org/index.html). For exon alignment, all transcript isoforms of one gene were compared both intra-species (within human) and inter-species (between human and the great apes) in order to confirm the specificity of the new transcript in humans. The human principal isoform, the human variant isoform, the chimpanzee isoform, the gorilla isoform, and the orangutan isoform of the *PCCB* gene are ENST00000469217 (NM_001178014), ENST00000466072, ENSPTRT00000028811, ENSGGOT00000005484 and ENSPPYT00000016431, respectively. The human principal isoform, the human variant isoform, the chimpanzee isoform, the gorilla isoform, and the orangutan isoform of the *STAU1* gene are ENST00000371856, ENST00000340954, ENSPTRT00000050938, ENSGGOT00000048652 and ENSPPYT00000012903, respectively.

The analysis for protein domain was performed using the Simple Modular Architecture Research Tool (SMART) in the normal mode (http://smart.embl-heidelberg.de). The protein accession numbers in humans, chimpanzees, gorillas, and orangutans for KMT2D are NP_003473, XP_016778992, XP_018894141, XP_024112209, respectively. The protein accession numbers in human, chimpanzee, gorilla, and orangutan for TRIOBP are NP_001034230, XP_016794633, XP_004063488, and H2P4B5 (UniProt), respectively. We performed multiple alignments using the software Constraint-based Multiple Alignment Tool (COBALT, https://www.ncbi.nlm.nih.gov/tools/cobalt/re_cobalt.cgi).

### Expression analysis for the candidates for intelligence evolution

We assessed the expression patterns of the HIEGs using transcriptome data for humans and their closest relatives. The HPA RNA-seq data were downloaded from the Human Protein Atlas (http://www.proteinatlas.org), including 102 samples of 37 tissues, in which TPM (transcripts per million) were used for the evaluation of expression level. In order to compare the expression levels between humans and great apes, the RNA-Seq data (NCBI ID: (100796) from 107 samples in 8 brain regions of humans, chimpanzees, gorillas, and gibbons were used (*Xu et al., 2018*). The brain tissues included five neocortical areas and three other brain tissues. The five neocortical areas are the anterior cingulate cortex (ACC), the dorsolateral prefrontal cortex (DPFC), the ventrolateral prefrontal cortex (VPFC), the premotor cortex (PMC), the primary visual cortex (V1C). The three other brain tissues are the hippocampus (HIP), the striatum, and the cerebellum (CB). Hierarchical cluster analysis was applied using the RPKM (reads per kilobase per million) of 39 genes in 8 brain tissues of humans and the other three primates, and was displayed using the software MeV4.2 (http://mev.tm4.org/).

## RESULTS

### Identification of candidate genes for human intelligence evolution

The strategy that was used for detecting the candidate genes in the evolution of human intelligence is briefly described (Fig. 1). Hundreds of genetic loci for human intelligence and its related traits have been identified by GWAS using large population data. We integrated six high-quality GWAS works and large-scale GWAS meta-analyses from the last five years. This enabled the identification of a total of 271 loci in the human

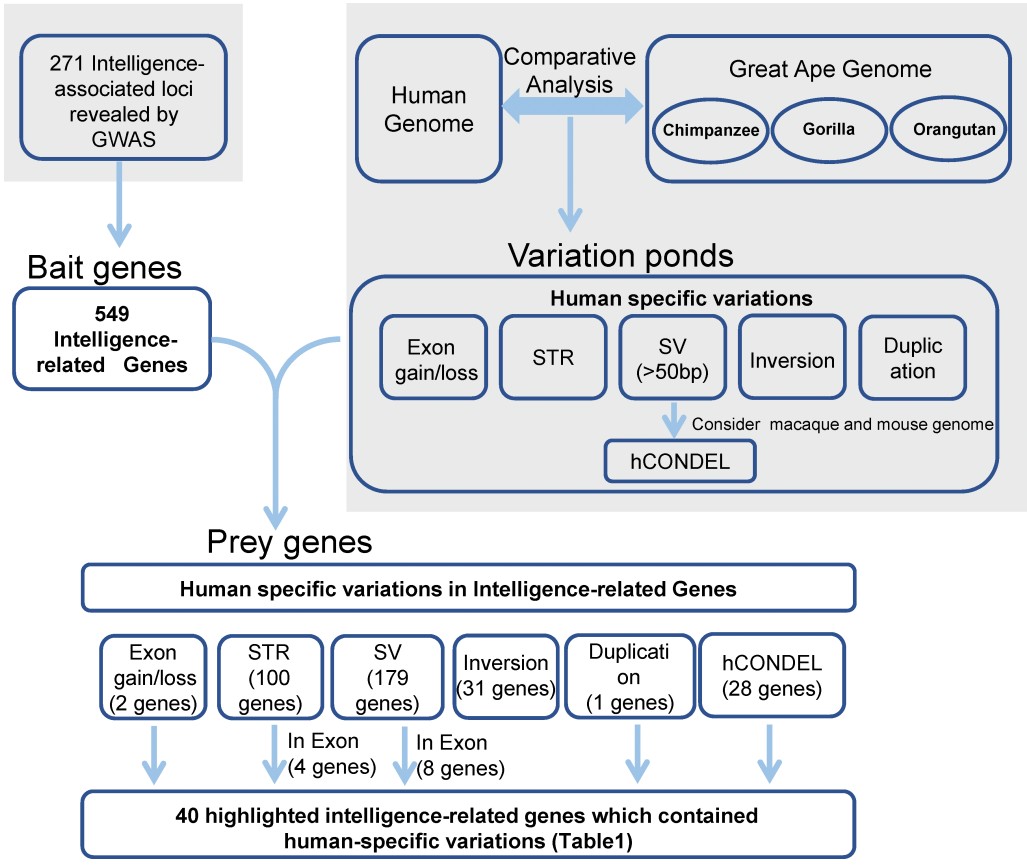

**Figure 1  Strategy for searching for genes in human intelligence evolution.** Grey areas indicated results from the previous studies including six GWAS works on intelligence and the comparative genomics analysis between humans and great apes.

genome underlying intelligence-related phenotypes (Table S1). A total of 549 human genes, distributed across the whole human genome, were found to be located within the 271 intelligence-associated loci (Fig. 2, Table S2). We take the 549 genes as "bait genes", which constituted intelligence-related genes from GWAS, underlying the intra-species intelligence-related variations related to intelligence in human populations.

In the meantime, the recently released long-read sequence assembly of chimpanzee, orangutan, and gorilla provided a large number of high-quality sequence differences between the human genome and the great ape genomes. We incorporated the human-specific structural variations, including exon gain and loss, STR, indels, hCONDEL, HSDs, and large structural variations, into a "variation pond" which are related to human evolution.

The "bait genes" and "variation pond" were then integrated to detect whether there were "bait genes" hit by the human-specific variation. After putting the "bait genes" into the "variation pond", we found 406 sequence variations physically located within 213 genes related to intelligence, which is considered to be "prey genes" (Fig. 2). The 213 "prey genes" linked the inter-species and intra-species variations and may be related to the

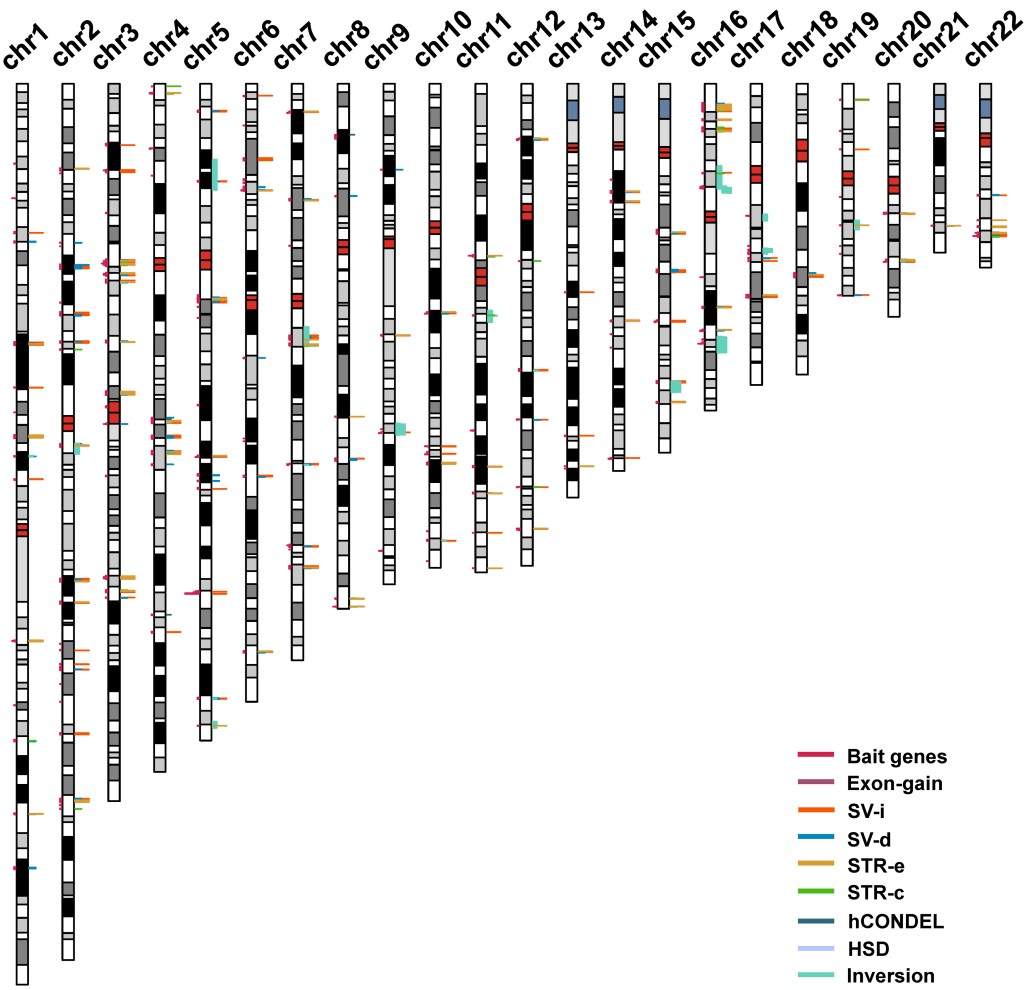

**Figure 2** The distribution of intelligence-associated genes and the human-specific variations which were located in them in the 22 autosomes of the human genome. The intelligence-associated genes are indicated on the left side of the chromosome bars. The human-specific variations around the intelligence-associated loci are indicated by lines of colors on the right side of chromosome bars. The centromere regions are indicated by red boxes. STR-c: STR contraction; STR-e: STR expansion; SV-d: deletions; SV-i: insertions; HSD: human segmental duplications; and hCONDEL: human conserved deletions.

evolution of human intelligence. Additionally, the potential effects of the human-specific variation through local sequence comparisons and gene structure analyses were carefully checked. Following in-depth analyses, there were 40 strong candidate genes that were identified as containing human-specific variations, probably changing either the coding or the expression of intelligence-related genes, which were named as human intelligence evolution genes (HIEGs, Table 1).

## Exon gain and loss on the intelligence-related genes

There were only two genes classified as the "prey genes" with exon gain or loss events, *PCCB* and *STAU1* (Table 1). The *PCCB* gene encoding the propionyl-CoA carboxylase subunit beta, was located in the locus on chromosome 3 identified by GWAS for intelligence
**Table 1** The highlighted human intelligence evolution-related candidates (HIEGs).

| Gene | CHR | Gene loci | Related SNP | Variation type | Variation position | Variation length | Genetic disorders |
|---|---|---|---|---|---|---|---|
| KDM4A | 1 | 43650158-43705515 | rs2842188[a] | hCONDEL | 43656932 | 466 | |
| NRXN1 | 2 | 49918505-51225575 | rs7557525[a] | hCONDEL | 50146470 | 3405 | Pitt Hopkins like syndrome2 |
| | | | | hCONDEL | 50146625 | 958 | |
| | | | | insertion | 50827590-50829782 | 2193 | |
| AFF3 | 2 | 99545419-100142739 | rs71413877[a] | HSD | 100080237-100104859 | 24623 | |
| | | | rs13010010[b] | hCONDEL | 99554281 | 1250 | |
| THSD7B | 2 | 136765545-137677717 | rs2558096[a] | hCONDEL | 137333693 | 1212 | |
| ARIH2 | 3 | 48918821-48986382 | rs13096357 | insertion (STR expansion)[*] | 48968205-48968380 | 176 | |
| | | | rs2352974[a] | | | | |
| | | | rs73078367[a] | | | | |
| STAB1 | 3 | 52495338-52524495 | rs4687625[a] | STR expansion | 52509368-52509454 | 87 | |
| SFMBT1 | 3 | 52903572-53046750 | rs4687625[a] | hCONDEL | 52951656 | 4034 | |
| FOXP1 | 3 | 70952817-71583993 | rs11720523[a] | hCONDEL | 71460722 | 449 | Mental retardation |
| | | | | hCONDEL | 71162689 | 1760 | |
| CADM2 | 3 | 84958981-86074429 | rs6770622[a] | hCONDEL | 85947229 | 1042 | |
| PCCB | 3 | 136250306-136337896 | rs9853960[a] | exon_gain | 136326325-136326385 | 60 | Propionic acidemia |
| TFDP2 | 3 | 141944428-142149544 | rs10804681[a] | hCONDEL | 142144111 | 2011 | |
| GRID2 | 4 | 92303622-93810157 | rs1972860[a] | hCONDEL | 92649012 | 135 | Spinocerebellar ataxia |
| BANK1 | 4 | 101411286-102074812 | rs13107325[a] | hCONDEL | 101422878 | 3650 | |
| | | | | hCONDEL | 101990081 | 4838 | |
| TTC29 | 4 | 146706638-146945882 | rs6840804[a] | hCONDEL | 146796204 | 9671 | |
| PDE4D | 5 | 58969038-60522120 | rs34426618[a] | insertion | 60429841-60432132 | 2292 | |
| PAM | 5 | 102753981-103031105 | rs76160968[a] | hCONDEL | 102883977 | 3868 | |
| FBXL17 | 5 | 107859035-108382098 | rs1438660[a] | hCONDEL | 108119106 | 959 | |
| | | | rs12187824[a] | | | | |
| CALN1 | 7 | 71779491-72447151 | rs56150095[a] | hCONDEL | 72221700 | 2994 | |
| SND1 | 7 | 127652180-128092609 | rs4731392[a] | hCONDEL | 127808446 | 749 | |
| EXOC4 | 7 | 133253073-134066589 | rs1362739[b] | insertion | 133889352-133895456 | 6105 | |
| | | | rs4728302[a] | | | | |
| SGCZ | 8 | 14089864-15238339 | rs13253386[a] | hCONDEL | 14090971 | 277 | |
| TSNARE1 | 8 | 142212080-142403240 | rs4976976[a] | STR expansion | 142326108-142326158 | 51 | |
| REEP3 | 10 | 63521363-63625123 | rs2393967[a] | hCONDEL | 63593088 | 554 | |

Li et al. (2020), *PeerJ*, DOI 10.7717/peerj.8912

**Table 1** (*continued*)

| Gene | CHR | Gene loci | Related SNP | Variation type | Variation position | Variation length | Genetic disorders |
|------|-----|-----------|-------------|----------------|--------------------|------------------|-------------------|
| GRIA4 | 11 | 105609994-105982092 | rs7116046[a] | hCONDEL | 105754761 | 3804 | Neurodevelopmental disorder |
| NCAM1 | 11 | 112961247-113278436 | rs2885208[a] | hCONDEL | 113152794 | 160 | |
| RERG | 12 | 15107783-15348675 | rs55754731[a] | hCONDEL | 15107134 | 69 | |
| KMT2D | 12 | 49018975-49059774 | rs1054442[a] rs146865992[a] | STR contraction | 49032866 | 60 | Kabuki syndrome |
| PRKD1 | 14 | 29576479-30191898 | rs971681[a] | hCONDEL | 29869703 | 1515 | Congenital heart defects and ectodermal dysplasia |
| FUT8 | 14 | 65410592-65744121 | 14:66113725 _C_A[c] | insertion | 65457887-65458201 | 315 | Glycosylation disorder |
| RTF1 | 15 | 41408408-41483563 | rs75322822[a] | hCONDEL | 41430384 | 1965 | Language delay and ADHD/ cognitive |
| GNB5 | 15 | 52115105- 52191369 | rs7172979[a] | hCONDEL | 52177036 | 1472 | |
| | | | | insertion | 52121612-52121903 | 292 | |
| SKAP1 | 17 | 48133440- 48430275 | rs12928404[b] | hCONDEL | 48286479 | 343 | |
| | | | | hCONDEL | 48259161 | 53 | |
| DCC | 18 | 52340172-53535903 | rs71367283[a] rs6508220[a] rs6508220[a] | hCONDEL | 52358208 | 566 | Corpus callosum agenesis |
| ZNF584 | 19 | 58401504-58418327 | rs73068339[a] | insertion | 58404219-58406377 | 2159 | |
| SLC27A5 | 19 | 58479512-58512413 | rs73068339[a] | deletion | 58490956 | 3235 | |
| PHF20 | 20 | 35771974-35950381 | rs78084033[a] | hCONDEL | 35797866 | 3808 | |
| STAU1 | 20 | 49113339-49188367 | rs6019535[a] | exon_gain | 49179121-49179244 | 124 | |
| DDX27 | 20 | 49219295-49244077 | rs6019535[a] | hCONDEL | 49221110 | 809 | |
| TRIOBP | 22 | 37696988-37776556 | rs4396807[a] | insertion | 37723443-37724117 | 675 | Nonsyndromic deafness |
| EP300 | 22 | 41091786-41180079 | rs4821995[a] | hCONDEL | 41135508 | 2279 | Rubinstein-Taybi syndrome2 |

**Notes.**

*The insertion was also identified as STR expansion.

[a]This locus was identified for intelligence in *Savage et al. (2018)*.

[b]This locus was identified for intelligence in *Sniekers et al. (2017)*.

[c]This locus was identified for intelligence in *Davies et al. (2016)*.
($P = 1.956 \times 10^{-9}$, *Savage et al., 2018*). Human *PCCB* contained 15 protein coding isoforms according to the Ensembl release 95. The principal isoforms generated proteins of 539 aa, 559 aa and 570 aa, respectively, all of which have identical isoforms in the chimpanzee. However, a new transcript variant was human-specific. Compared with the principal isoform, the variant lost a 60-bp exon (exon 4 of principal isoform) but gained another new 60-bp exon (exon 11 of the variational isoform). The gained 60-bp exon did not appear in any transcripts of the *PCCB* of the three great apes (Fig. 3A). This human-specific transcript variant can be detected in the human cerebral cortex at a lower level than that of the principal isoform (transcriptome data were obtained from the Human Protein Atlas, HPA). However, the relative cerebral cortex expression level of the *PCCB* variational transcript was higher than that of the principal transcript (relative to the average expression level of the transcript in 37 tissues) (Figs. 3B and 3C). These results suggested the generally low expression level and the relative enrichment of the *PCCB* variant in the cerebral cortex, implying a potential role of the human-specific *PCCB* transcript variant in the cerebral cortex. Although there were no reports about the mechanism of *PCCB* function in neuron system development, mutations in *PCCB* are one of the major causes of the genetic disease propionic academia (PA). Neurological complications, such as intellectual disability, brain structural abnormalities, optic neuropathy, and cranial nerve abnormalities are significant symptoms of PA (*Schreiber et al., 2012*). Moreover, there were reports that patients carrying *PCCB* mutations exhibited intellectual disabilities (*Witters et al., 2016*).

*STAU1* was also located in a locus identified by GWAS for intelligence (*Savage et al., 2018*). *STAU1* encoding the double-stranded RNA-binding protein which regulates RNA metabolism. There was a total of 10 protein-coding isoforms of human *STAU1*. Compared with the longest principal isoform, a human-specific insertion resulted in a gain of 123-bp exon (exon 2, located within the 5′UTR of the gene) in one transcript variant. The isoform, with the addition of a 123-bp exon, was not detected in any transcripts of chimpanzee, gorilla or orangutan (Fig. 3D). The human-specific isoform was expressed in many human tissues. In the human cerebral cortex, the expression of the new isoform was equivalent to ~21% of that of the principal isoform of the gene (Figs. 3E and 3F). Previous functional studies found that STAU1 plays a role on mRNA transport in neuronal dendrites (*Broadus, Fuerstenberg & Doe, 1998*).

## STR variations on the intelligence-related genes

In the variation pond, there were a total of 1,465 human-specific STR contractions and 4,921 human-specific STR expansions. These STRs hit 100 "bait genes" (26 genes containing the STR contractions and 74 genes containing the STR expansions, Table S3). Most of the human-specific STR variations were located within the intron regions or intergenic regions. Only four human-specific STRs were located within the exonic regions, highlighted as HIEGs (Table 1). Among them, three STR expansions were located within the exon of non-protein coding isoforms (processed transcripts) of three genes (*ARIH2*, *STAB1* and *TSNARE1*), and one STR contraction was located within the 39th exon of the *KMT2D* gene.

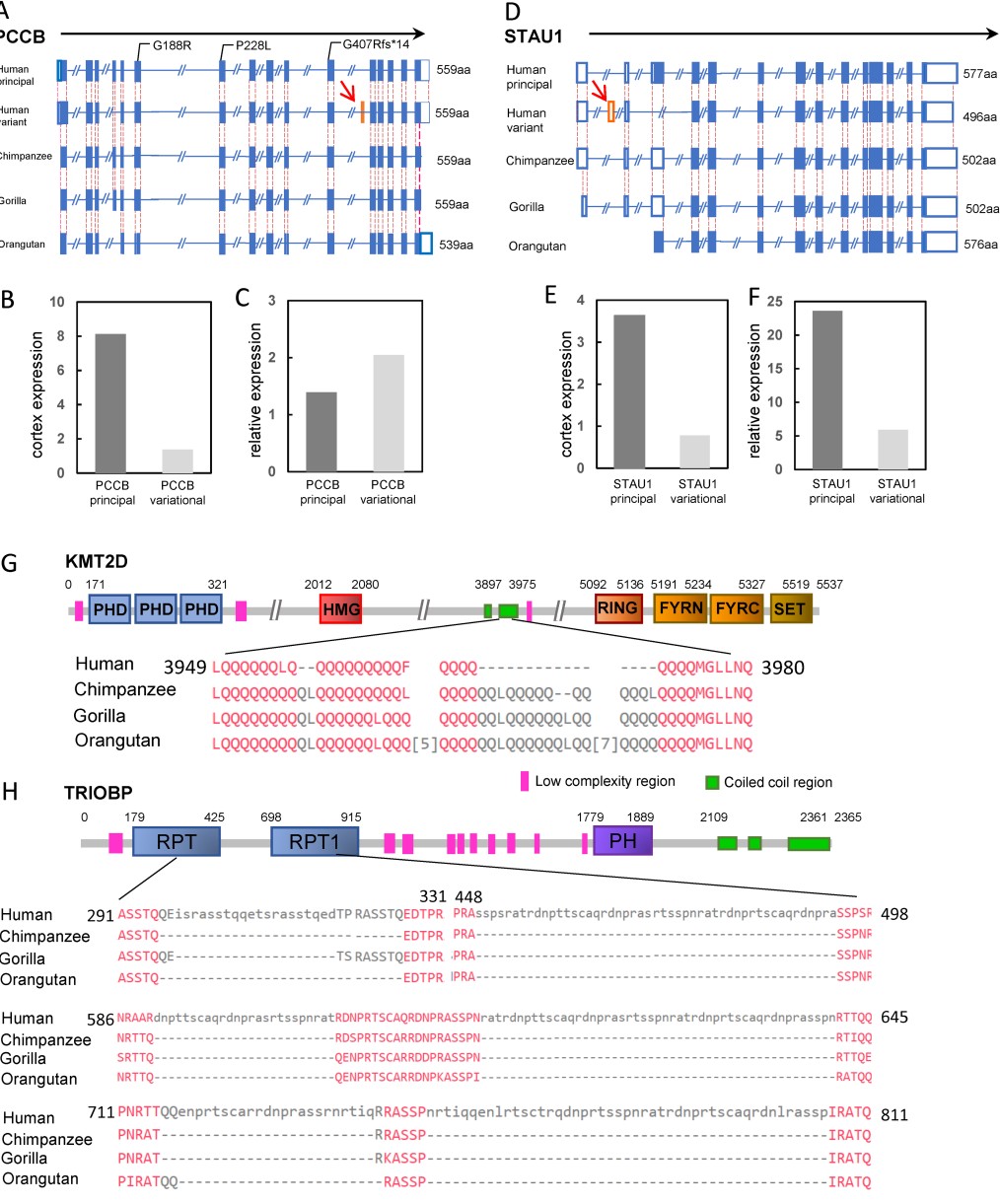

**Figure 3  Four candidate genes involved in human intelligence evolution.** (A) Gene structures of PCCB in human and great ape genomes. The orange box represents the gained exon (also indicated by red arrow). The mutations in patients with intellectual disability are labeled. The solid boxes represent coding exons, and the hollow box represent UTRs; (B) TPM of PCCB principal isoform and variant isoform in cerebral cortex; (C) TPM of PCCB principal isoform and variant isoform in cerebral cortex relative to 37 human tissues; (D) Gene structures of STAU1; (E) TPM of STAU1 principal isoform and variant isoform in cerebral cortex; (F) TPM ofSTAU1 principal isoform and variant isoform in cerebral cortex relative to 37 human tissues; (G) The schematic representation of KMT2D protein; (H) The schematic representation of TRIOBP protein. The protein sequence alignments of regions with human-specific variation are showed. The numbers indicated the position of amino acids, and the numbers in square brackets between sequences indicated the hided amino acids.

*KMT2D*, also known as *MLL2*, encoded an H3K4 histone methyltransferase made up of 5537 amino acids. GWAS for intelligence detected one associated locus ($P = 2.518 \times 10^{-14}$, *Savage et al., 2018*) on chromosome 12 containing the candidate gene *KMT2D*. Compared with the *KMT2D* sequence in the great apes, the STR region contracted for 60-bp in the c.11838 of human *KMT2D*. This region was also polymorphic among the human population. The contraction led to a 16-aa discontinuous deletion for the human KMT2D protein from the p.3958 position (Fig. 3G). The peptide coding by this STR region was in the coiled coil region for KMT2D, which could affect the protein structure through wrapping the hydrophobic residues and forming an amphipathic surface (*Mason & Arndt, 2004*). Mutations in *KMT2D* are the main cause of the genetic disease Kabuki syndrome, and several mutations in the 39th exon have been found in Kabuki patients. Kabuki syndrome affected mental capabilities and most of these patients showed various levels of intellectual disability (*Lehman et al., 2017*).

## Indels on the intelligence-related genes

Among 5,894 human-specific deletions and 11,899 human-specific insertions in the "variation pond", we found 94 "prey genes" with 148 deletions and 144 genes with 298 insertions (Table S4). Furthermore, it was found that there were 8 insertions and 1 deletion for the exonic regions, as highlighted by the HIEGs (Table 1). The insertion in *ARIH2* was also identified as STR expansion which has been described above. Insertions in *PDE4D*, *NRXN1*, *EXOC4*, *FUT8*, and *ZNF584*, and the deletion in *SLC27A5* affect the lengths of the non-protein coding isoforms (processed transcripts) of the six genes. Moreover, *GNB5* and *TRIOBP* carried an insertion in the 3′UTR region and an insertion in the exon region, respectively.

The insertion in *TRIOBP* resulted in a gain of 675-bp coding region (c.887-1560) in the 5th exon of the longest isoform when compared with that in the chimpanzee. This variation resulted in a 238-aa discontinuous insertion in the region p.296–811 of the human TRIOBP protein (Fig. 3H). The GWAS for intelligence ($P = 3.582 \times 10^{-8}$, *Savage et al., 2018*) and the GWAS for underlying brain ventricular volume also identified the gene *TRIOBP* as a strong candidate (*Vojinovic et al., 2018*). Biochemistry experiments have shown that TRIOBP could physically interact with TRIO, which is an important protein involved in neural tissue development (*Seipel et al., 2001*). Mutations in *TRIOBP* could cause autosomal recessive deafness-28 (DFNB28), and surprisingly, several causal mutations were located within the human-specific insertion regions (e.g., R347X and Q297X) (*Shahin et al., 2006*).

We also searched for human conserved deletions (hCONDELs) near the "bait genes" that had been identified (*McLean et al., 2011*; *Kronenberg et al., 2018*). These sequences are lost in the human genome but are highly conserved among other species (including the great apes, the macaque, and the mouse). In total, 28 genes were identified as containing the hCONDELs. Although none of the 28 hCONDELs were located in the coding regions, they were all taken into HIEGs (Table 1), as the high lineage specificity of hCONDELs serving as important signs of intelligence evolution. The list included *NRXN1*, *GRID2* and *GRIA4*, which were all involved in neurotransmission and the formation of synaptic contacts.

The *GNB5* gene in the intelligence-associated locus on chromosome 15 ($P = 2.47 \times 10^{-11}$, (*Savage et al., 2018*)), which encoded the beta subunit of the heterotrimeric GTP-binding proteins (G proteins). Aligned with the *GNB5* sequences of the humans and the great apes, we found that there was a 292-bp human-specific insertion in the 3′UTR of the gene and a 1,472-bp hCONDEL in the intron (2.7 kb distance to the third exon) of the gene (Table 1). *GNB5* was expressed in the brain tissues and participated in neurotransmitter signaling (e.g., through the dopamine D2 receptor) (*Xie et al., 2012*). In human populations, mutations in *GNB5* have been reported to cause several diseases affecting intelligence, including language delay, ADHD/cognitive impairment with or without cardiac arrhythmia, and intellectual developmental disorder with cardiac arrhythmia (*Lodder et al., 2016*). The human-specific variation in GNB5 might participate in the evolution of human intelligence as well.

## Large structural variations on the intelligence-related genes

The inversion variation is the rearrangement in which the genomic segment is reversed. Based on the previous report, there were a total of 625 inversions in our "variation pond" ranging from 9 kb to 8.4 Mb in size. Among these, 31 of them hit the "bait genes" (Table S5). However, none of these genes were located in the breakpoint of the human-specific inversions.

There were 218 human-specific duplications (HSD) the length of which were more than 5-kb (*Prado-Martinez et al., 2013*). Among them, one 24.6-kb HSD was detected to be overlapped with the *AFF3* gene. The *AFF3* also contained an hCONDEL around the intron-exon junction regions (31-bp distance to the exon). The region around *AFF3* has been identified to be an intelligence-associated locus in found in two distinct meta-analyses conducted on intelligence ($P = 1.56 \times 10^{-8}$, *Sniekers et al., 2017* or $P = 3.41 \times 10^{-10}$, *Savage et al., 2018*), but no functional reports on neuron development are available.

## Expression profiling analysis of the HIEGs

The transcriptome data of 37 human tissues from the Human Protein Atlas database (*Uhlen et al., 2015*) was used to investigate the tissue expression patterns of the 40 highlighted HIEGs (Fig. 4A). Of these there were 23 genes with higher expression levels in the cerebral cortex than their average expression levels in all 37 tissues. Furthermore, there were 12 genes with more than twofold expression levels in the cerebral cortex versus their average expression levels (Fig. 4A). These were regarded as genes that were potentially involved in the development of the cerebral cortex. Except *FUT8*, the remaining 11 genes are the hCONDEL-containing genes. The expression patterns of the 11 genes exhibited three types. Four genes, including *GRIA4*, *NRXN1*, *CADM*2, and *CALN,* showed the most significant high expressions in the cerebral cortex and low expressions in the other tissues. *SGCZ*, *DCC,* and *GRID2* showed only relatively high expressions in the cerebral cortex compared with the low expressions in all other tissues. *NCAM1*, *FBXL17*, *FUT8* and *GNB5* were generally expressed in all tissues (Fig. 4A).

A transcriptome dataset sampling eight brain regions (five neocortical areas, hippocampus, striatum, and cerebellum) of both humans and four primate species

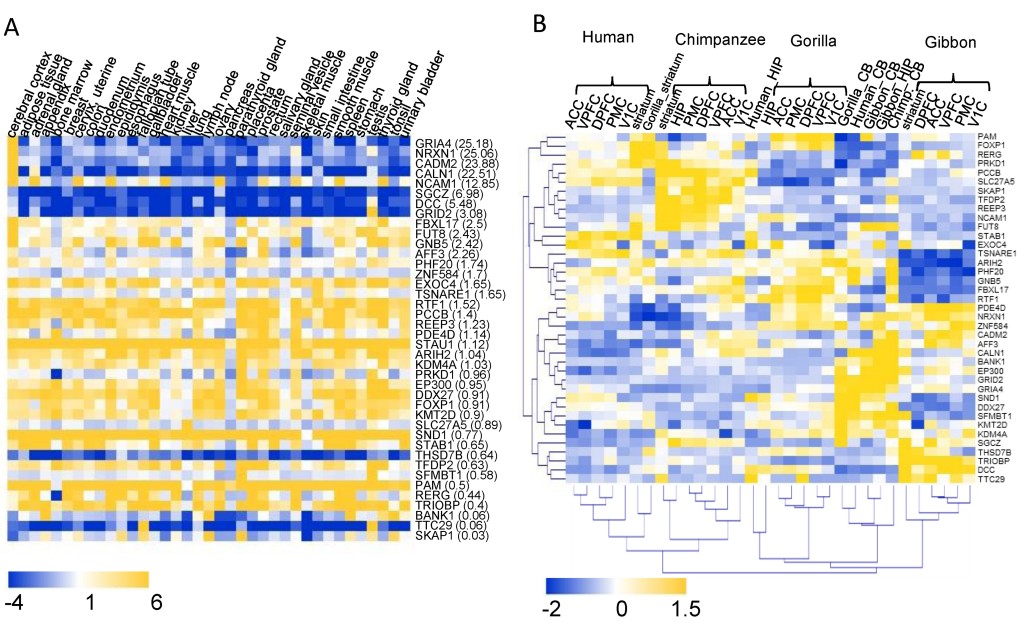

**Figure 4 Expression profiling of the 40 HIEGs.** (A) The expression levels of the 40 genes in 37 human tissues (data from the HPA). The sort order of the genes from top to bottom is based on the ratio of the expression in cortex relative to the average in 37 tissues, which is indicated behind each gene name. (B) Hierarchical clustering of the expressions of 39 HIGEs in the 8 brain tissues of human, chimpanzee, gorilla, and gibbon. ACC: anterior cingulate cortex; DPFC: dorsolateral prefrontal cortex; VPFC: ventrolateral prefrontal cortex; PMC: premotor cortex; V1C: primary visual cortex; HIP: hippocampus; striatum; CB: cerebellum.

(chimpanzees, gorillas, gibbon, and macaques) was reported previously (*Xu et al., 2018*), which enabled a comprehensive interspecies comparison. This dataset was used to examine whether the genes containing human-specific variation showed expression changes in the cortex between humans and the great apes. The expression data for 39 of the 40 HIEGs, with the exception of *STAU1,* could be found in the transcriptome dataset. Among them, the expression levels of the genes *AFF3, SKAP1, REEP3, DCC,* and *SGCZ* in the human neocortical areas were much lower than those in the neocortical areas of the great apes (fold change < 0.5), while the expressions of *STAB1* in the human neocortex were much higher than those in the great apes (fold change = 4.8). Hierarchical clustering was also performed for the 39 gene expressions in all samples. We found that 5 neocortex tissues in the same species could be always clustered (that is, one clade for one species), while CB, STR, and HIP were generally clustered based on their tissue types (Fig4 B). This result suggested that the expression profiles of the HIEGs in the neocortex tissues displayed a strong species specificity, which was in contrast to the profiles in these non-neocortex tissues (e.g., CB). Taken together, the intelligence-related genes, containing the human-specific variations, expressed differently in the neocortex tissues between the humans and the great apes.

## DISCUSSION

The understanding of human intelligence from the view of evolutionary genetics is an important scientific undertaking. Science magazine posted 125 scientific questions which were driving basic scientific research in various disciplines (*Kennedy & Norman, 2005*); among the top 25 questions, human evolution was addressed with the question: "What Genetic Changes Made Us Uniquely Human?" (*Culotta, 2005*). Obviously, genetic changes related to intelligence is one of the key steps in human evolution, making us uniquely *Homo sapiens*. However, it is still very challenging to identify the causative genes that are responsible for the intelligence difference between human and the other species in Hominidae. With both the human genome sequence and those of great apes becoming available, it is possible to pinpoint the genetic changes underlying the phenotypic differences between humans and the great apes. A subset of genes that were related to the human-ape intelligence differences may also contain intraspecific allelic variation underlying the variation of intelligence levels in human populations. Hence, in this work we integrated both the latest GWAS information on intelligence and the latest advances in the great ape genomics, aiming to mine the gene clues to understand the evolution of human intelligence. We found several strong candidates, for example, the genes *TRIOBP* and *GNB5* contain human-specific variations and have the genetic evidence to be involved in the development of intelligence (*Vojinovic et al., 2018*; *Xie et al., 2012*; *Lodder et al., 2016*), although more in-depth molecular evidence and validations are needed in future experiments. Although our work did find some candidate genes involved in human intelligence evolution, this strategy has its shortcomings. Some of intelligence-related genes with inter-species (between human and the great apes) variation may contain no intra-specific variation in human populations and cannot be identified by GWAS. Hence, the human genes involved in intelligence evolution but without intra-species variations in human populations were missed using our strategy.

We identified a total of 40 HIEGs, based on the location of the human-specific variations. Except for the hCONDELs as their high lineage specificity and importance in evolution, all the other human-specific variations in the HIEGs were located within the exons or putative regulatory regions. However, some of them were located within the exon of non-protein coding isoforms. There were five genes carried human-specific variations in exons or UTRs of protein coding isoforms. In this paper only five HIEGs were addressed. It was notable that two of the 5 genes were related to mRNA decay. One candidate gene with an exon gain, Staufen1 (*STAU1*), were reported to be involved in the transport, relocation, translation of mRNA and mRNA decay (*Paul et al., 2018*). Loss of *STAU1* function in mice resulted in impaired mRNA transport and reduced synapse formation (*Vessey et al., 2008*). Another candidate gene, *KMT2D* harbored a human-specific STR contraction. Several truncating mutations within *KMT2D* resulting in mRNA degradation through nonsense-mediated mRNA decay, contributing to protein haploinsufficiency (*Micale et al., 2014*). It is still unclear whether there are any functional links between the two genes for mRNA processes in brain developments.

The associated loci from GWAS in humans often contain several candidate genes due to genetic mapping resolution, which is one of the difficulties in our bioinformatics analyses. To avoid artificial bias, all the candidate genes around the associated loci were included in the collection of "bait genes", although usually for each associated locus only a single gene is causative. Consequently, a total of 549 human genes were included for the 271 intelligence-associated loci and "bait genes" contain many false positives. We cannot clearly distinguish the true one with the highly linked one, because other information (e.g., based on expression profiles or the distance to lead SNPs) is often misleading. As a result, HIEGs must contain many unrelated genes, although the intelligence-related genes involved in human evolution have been partly enriched. Further experiments and analyses may include the validation of gene functions (whether and how these genes could influence the development of human brains) and the assessment of the effects of the human-specific variation (whether and how these sequence variations could influence the gene coding or the gene expression patterns).

There are already several findings of the human-chimpanzee differences altering the development of the neocortex to date. The knowledge from the works of gene functional studies and evolutionary genetics studies greatly enhanced our understanding of intelligence and the brain (*Boyd et al., 2015*; *Dennis et al., 2012*; *Charrier et al., 2012*; *Florio et al., 2015*; *Ju et al., 2016*). Certainly, the known genes (e.g., *NOTCH2NL*, *FZD8*, *SRGAP2*, *ARHGAP11B*, and *TBC1D3*) are only a small proportion of the whole gene set that encapsulates the vast differences in brain size and intelligence between great apes and humans, leaving many remaining gaps in our knowledge. More integrated approaches incorporating genetics, genomics, bioinformatics, and development biology will be needed in future works.

## CONCLUSION

GWAS has identified hundreds of genes associated with intelligence variation in human populations. Through inter-species genome comparisons between human and great apes, we found that a small proportion of intelligence-related genes also contained human-specific variation. Through integrated approaches, especially the careful checking of sequence alignments and gene annotations, we identified 40 candidates in which human-specific variation may have effects on gene coding or expressions. Transcriptome-wide comparison between humans and four primate species for the 40 candidate genes suggests that several of them displayed a different expression pattern among these species. The results implied that at least a few of the intelligence-related genes may contain both intra-species variation and inter-species variation. The intra-species variation underlies the small variation of intelligence levels among different human individuals while the inter-species variation controlled the large genetic differences of intelligence between the great apes and humans. This work may provide a list of candidate genes to be used in subsequent studies.

## ACKNOWLEDGEMENTS

We thank Professor Xuehui Huang from Shanghai Normal University for providing computer servers.

### Funding

This research was supported by the National Natural Science Foundation of China (81402212) and the Faculty Start-up Funds of Shanghai Normal University. The funders had no role in study design, data collection and analysis, decision to publish, or preparation of the manuscript.

### Grant Disclosures

The following grant information was disclosed by the authors:
The National Natural Science Foundation of China (81402212).
The Faculty Start-up Funds of Shanghai Normal University.

### Competing Interests

The authors declare there are no competing interests.

### Author Contributions

- Mengjie Li and Wenting Zhang performed the experiments, analyzed the data, prepared figures and/or tables, and approved the final draft.
- Xiaoyi Zhou conceived and designed the experiments, analyzed the data, prepared figures and/or tables, authored or reviewed drafts of the paper, and approved the final draft.

### Data Availability

The raw code is available as a Supplemental File.

### Supplemental Information

Supplemental information for this article can be found online at http://dx.doi.org/10.7717/peerj.8912#supplemental-information.

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
