# Peer review of "Identification of genes involved in the evolution of human intelligence through combination of inter-species and intra-species genetic variations"

_PeerJ, doi:10.7717/peerj.8912_

## Round 0.1 · original submission · Major Revisions

As you will see, both reviewers gave constructive and thoughtful comments, and I believe that the manuscript will be considerably improved with the incorporation of the suggested revisions. In particular, careful attention to the language of the revised manuscript as it related to the scientific meaning of the text. As such, please consider having the resubmitted version proofread by a trusted colleague in your field with native fluency in English. I am sensitive to the difficulty of publishing in a second (or third, or more) language, and I respect that effort, but it is also a fact that the clarity and reception of the final product are in part dependent on the quality of the written language.

Please identify the changes in your revision clearly, and if necessary, I will be happy to consider any additional information or rebuttals you would like to provide. Thank you again for submitting your work to PeerJ, and I look forward to seeing the next iteration of this manuscript.

·

Basic reporting

The language must be improved throughout the article. I provided a few suggestions covering all up to the results section, but ultimately reckoned this to be to daunting a task for me to address. I suspect that no person with English as mother tongue has had any part in writing this manuscript. I strongly advise the authors to submit it to a native English speaker with adequate scientific knowledge for proofreading before resubmission.
The article's context is otherwise relatively clearly stated and the literature references seem satisfactory. I annotated the text in the submission pdf whenever I felt a reference was missing.
The methods and results section intersect each other somewhat. I think the article would benefit from a more clear separation between method description and elucidation of results. The figures are mostly fine. I annotated the submission pdf wherever I found anything amiss with them. Figure 2 could perhaps benefit from higher resolution.

Experimental design

I think the study is original and in line with the aims and scope of the journal. The research question is clear. The aim of this study is quite ambitious. To what extent the highlighted knowledge gap is filled is arguable, but the effort made by the authors is commendable.
The analyses are well described and should be replicable. I think the results should be elucidated more systematically. The authors found 40 genes with intra-species variation and potential evolutionary salience but do not discuss all of them.

Validity of the findings

I think the conclusions of the study are commensurate to its results. The only reservations I have are about the title and the emphasis on the novelty of the method in the abstract. As far as I understood, the authors essentially searched and combined multiple databases. The comparative genomic analysis they performed served only the purpose of pruning existing (fixated) human-specific variation databases (by the way, how much does this pruning actually affect the results? is it necessary?). I think the title should be changed to highlight the combination of inter- and intra-species variation and de-emphasize the use of comparative genomics analysis. Also, the last statement of the abstract should be downplayed or possibly removed entirely.

Additional comments

I was particularly bothered by the ingenuous, Homo-centric language used by the authors throughout abstract and introduction. I feel that phrases such as “the evolution of human intelligence from great apes to humans” or “humans are much more intelligent than chimpanzees” do not belong in a scientific publication. It is common to use our chimpanzee cousins as evolutionary proxy of our common ancestor but it is a misconception to presume a temporal relationship between apes and humans, and one that should not be reiterated in a scientific work, even if only “in a manner of speaking”. Also, intelligence is a very hard concept to define and depends to a great extent on the context in which one attempts to measure it. In fact, it depends just as much on the context as it does on the subjects that are inserted in that context. The journal itself is called “Life & Environment”. One shouldn’t judge a fish by how skilled it is at climbing trees.
To sum up, I think the substance of the manuscript may remain as is, with only those few additions I indicated, but its form should be considerably improved. Make no mistake, I do consider the needed revisions major ones.

Reviewer 2 ·

Basic reporting

The authors have used clear and professional English in their article.
They have described in detail the various GWAS from which their genes baits are derived. They provide sufficient literature references. However, their work is not merely reporting their findings but they also claim that “the work provides a new method in searching for the key genes in human evolutionary genetics.” In such a case, I would suggest the authors give more background and write about other methods that are currently used in the field and compare to theirs. This will provide a better context of the strength or weakness of their method. I feel the authors have moved rather abruptly to specific information about the genes in line 46-47 and then the authors go back to describing other studies in line 58 onwards. Here, I would suggest writing a bit about the methods and shortcomings of those studies, not just their findings.
In line 72-75 the authors claim the previous reference genomes are not qualified enough but do not expand on why they are not qualified.
Minor point: I was not able to find reference to Figure 2 anywhere in the text. Some typographic errors such as spacing between words in the figure legends 1, 2 & 3.

Experimental design

The investigation seems to be of high standard but again, I had some doubts over the research questions and if they had been answered well. Was the aim to find genes associated with the evolution of intelligence or to devise a new method of finding such genes? Either way, the authors should place their results within the larger context of the field and discuss the strength and weakness of their method.

Validity of the findings

no comment

Additional comments

The authors have utilized a novel method to provide a list of candidate genes associated with evolution of intelligence in humans. While the genetic results are interesting, it would be helpful if the authors could discuss the potential weakness of their method and how they have built upon existing methods. The manuscript could improve with some elucidation and clarification about the method.

---

## Round 0.2 · Minor Revisions

I believe you paper has been considerably improved in revision, and with the incorporation of the remaining Reviewer-suggested changes, it will be improved even further. Reviewer 1 has kindly provided an annotated document with suggested changes to the language of the paper, and I hope you will consider using this as a foundation for your revisions. Please identify in your re-submitted document where you have addressed the Reviewer's suggestions, or provide a detailed explanation if you disagree with their comments. I look forward to seeing the next version, and I thank you again for submitting your work to PeerJ

·

Basic reporting

The authors have revised all instances of ingenuous or Homo-centric language, which were my major concern in their first draft. The language, however, still needs work. The authors have corrected a few but not all errors and have introduced new ones. I have applied some edits up to the results section in an annotated pdf but do not think the article's form meets the standards for a publication yet. I therefore reiterate my recommendation to submit the manuscript to a native English speaker for proofreading.
A couple of additional issues remain. They are commented on in the annotated pdf.

Experimental design

no comment.

Validity of the findings

no comment.

Additional comments

The descriptions of some of the submitted files appear with typos in the reviewer's dashboard. I'm not sure this is something the authors or the journal editor should fix but it would be best if it was not propagated.

Reviewer 2 ·

Basic reporting

The authors have improved the language and flow of the manuscript. The authors have provided more background and context after revision.

Experimental design

no comment

Validity of the findings

no comment

Additional comments

The authors have given more context and elucidated the points that were asked of them and the language is much better now.

---

## Round 0.3 · Minor Revisions

While I recognize the difficulty and increased burden of publishing in a non-native language, the current manuscript still has some language issues which need to be resolved before it could be published. The area of most concern is in the use of the classification terms hominid, human, great ape, etc. In several instances, "hominid" is used to refer to (apparently) humans, exclusive of great apes, whereas the modern term in this instance should be "hominin", as the taxon Hominidae includes the apes, except gibbons. In other places, "hominid" appears to refer to some, but not all apes, as well as humans (such as in "hominid-human comparative genomic analysis"; p.1), and in other places, the term "human" is used. Usage needs to be modified and standardized throughout the paper, and clear reference made to the groups that are being screened in each case.

The second remaining issue is the concept of human evolution as somehow separate from evolution in "nature", e.g. "...humans compete with nature..."; p.1 line 36. This sets up a conflict which is not relevant for the paper, and which many readers may find problematic. This is especially true as the authors, in the same sentence, describe humans competing with nature over "the past few million years", which is vastly longer than the existence of modern humans, to say nothing of civilization as it is currently recognized. This issue further compounds the problems associated with hominin/hominoid/human usage called out above. While this could be seen as purely a semantic issue, but given the topic of the paper and the difficulties of identifying what is "intelligence", what is "human", etc., they are of critical importance to the overall interpretability of the paper, and to identification of potential genes which could have been modified as features associated with "intelligence" appeared in the primate order.

I am therefore returning the paper for revision of these language issues, as well as some minor usage issues such as "Inspection on" vs. "Inspection of", which appear throughout the manuscript. Most readers would find the latter usage more accessible. Once these issues are resolved, I believe this work will make a substantive contribution to the literature on this complex topic.

---

## Round 0.4 · accepted · Accept

Thank you for your careful attention to the suggestions from the reviewers. I hope you will agree that their contributions have been helpful, and that you will be pleased with this outcome. I look forward to seeing the final published article. Congratulations, and thank you again for submitting your work to PeerJ.